# Effect of Surface Roughness and Electroless Ni–P Plating on the Bonding Strength of Bi–Te-based Thermoelectric Modules

**Sung Hwa Bae** [1] , **Sungsoon Kim** [1] , **Seong Hoon Yi** [1] , **Injoon Son** [1,*] , **Kyung Tae Kim** [2,*] **and Hoyong Chung** [3]

1   School of Materials Science and Engineering, Kyungpook National University, 80 Daehakro, Buk-gu, Daegu 41566, Korea; sy123421@knu.ac.kr (S.H.B.); tjdtns122@naver.com (S.K.); yish@knu.ac.kr (S.H.Y.)
2   Powder & Ceramic Materials Division, Korea Institute of Materials Science, 797 Chanwon-daero, Seongsan-gu, Changwon, Gyeongnam 51508, Korea
3   Department of Chemical and Biomedical Engineering, Florida State University, 2525 Pottsdamer Street, Building A, Suite A131, Tallahassee, FL 32312, USA; hchung@eng.famu.fsu.edu
*   Correspondence: ijson@knu.ac.kr (I.S.); ktkim@kims.re.kr (K.T.K.); Tel.: +82-53-950-5563 (I.S.); +82-55-280-3506 (K.T.K.); Fax: +82-53-950-6559 (I.S.); +82-55-280-3289 (K.T.K.)

**Abstract:** In this study, electroless-plating of a nickel-phosphor (Ni–P) thin film on surface-controlled thermoelectric elements was developed to significantly increase the bonding strength between Bi–Te materials and copper (Cu) electrodes in thermoelectric modules. Without electroless Ni–P plating, the effect of surface roughness on the bonding strength was negligible. Brittle SnTe intermetallic compounds were formed at the bonding interface of the thermoelectric elements and defects such as pores were generated at the bonding interface owing to poor wettability with the solder. However, defects were not present at the bonding interface of the specimen subjected to electroless Ni–P plating, and the electroless Ni–P plating layer acted as a diffusion barrier toward Sn and Te. The bonding strength was higher when the specimen was subjected to Ni–P plating compared with that without Ni–P plating, and it improved with increasing surface roughness. As electroless Ni–P plating improved the wettability with molten solder, the increase in bonding strength was attributed to the formation of a thicker solder reaction layer below the bonding interface owing to an increase in the bonding interface with the solder at higher surface roughness.

**Keywords:** Bi–Te thermoelectric; electroless Ni–P plating; bonding strength; surface roughness

## 1. Introduction

Owing to their high performance at temperatures below 200 °C, Bi–Te-based thermoelectric materials have been widely applied for active cooling using the Peltier effect and for power generation using the Seebeck effect [1–11]. The thermoelectric phenomena can be realized by fabricating thermoelectric devices/modules in which several tens to hundreds of n-type and p-type rectangular thermoelectric elements are electrically bonded in series on top of a copper electrode formed on the ceramic substrate [12]. Thus, for practical applications, the reduction of bonding failure has become crucial for achieving reliable thermoelectric performance of these modules.

In general, solder materials using Sn–Ag–Cu-based alloy, which has a melting temperature of approximately 220 °C, have been widely used for bonding Bi–Te-based thermoelectric elements and copper electrodes. However, Sn-based solder and Te materials generate brittle Sn–Te-based intermetallic compounds at a temperature of approximately 250 °C [13–17]. Many studies have reported that these Sn–Te-based intermetallic compounds deteriorate the bonding strength of

thermoelectric modules [12–16]. To improve bonding strength, a diffusion barrier between the thermoelectric element and solder is required.

The most widely applied methods are the deposition of a Ni or Ni alloy layer on the surface of the thermoelectric element [18–22]. That is, an Ni or Ni alloy acting as a diffusion barrier is incorporated by electroplating, electroless plating, physical vapor deposition (PVD), or the thermal spraying process. Among these methods, electroless Ni–P plating is promising, owing to its low cost and good processability in a uniform plating layer on the surface of the thermoelectric element via a simple dipping process. However, there are theoretical difficulties in achieving good adhesion because of the absence of chemical bonding between the Bi–Te-based alloys and the Ni–P layer.

In this study, to improve the adhesion between the thermoelectric element and the plating layer, we introduced a rough interface via mechanical treatment such as sand blasting to utilize the anchoring effect [23]. The surface roughness of the Bi–Te-based thermoelectric element was controlled via the sand-blasting method using alumina powder. Furthermore, the influence of Ni–P plated Bi–Te elements on the bonding strength of the thermoelectric module was investigated by changing surface roughness. Subsequently, cross-sectional observations at the interface and the measurement of the contact angle with the solder were carried out to determine the factors affecting the bonding strength.

## 2. Experimental Methods

Commercial n-type ($Bi_2Te_{2.7}Se_{0.3}$) and p-type ($Bi_{0.5}Sb_{1.5}Te_3$) Bi–Te alloy ingots (Daeyang Co., Ltd., Daegu, Korea) were crushed, and spark plasma sintering was used to prepare a sintered thermoelectric specimen. The sintered thermoelectric specimen was cut into a disk shape with a thickness of 3 mm via electric discharge machining. Three kinds of commercial-grade $Al_2O_3$ powders—#80 (mean particles size: 177 µm), #200 (mean particles size: 74 µm), and #400 (mean particles size: 37 µm)—were sprayed onto the disk surface using the sand blasting method to prepare thermoelectric elements with different surface roughness characteristics.

For electroless Ni–P plating pretreatment, the surfaces of the thermoelectric elements were etched via immersion in NaOH solution at 25 °C for 60 s, followed by immersion in a commercial Pd catalyst solution (ATOTECH, Adhemax Activator SF, Berlin, Germany) at 25 °C for 60 s to generate a Pd nucleation seed. Subsequently, the substrate was immersed in a commercial electroless Ni–P plating solution (YoungIn Plachem Co., Ltd., ENF, Ansan, Korea) at 88 °C for 20 min to produce a Ni–P plating layer approximately 3 µm thick on the surface of the Bi–Te-based thermoelectric element.

Subsequently, the disk-shaped thermoelectric element was cut into $3 \times 3 \times 3$ mm$^3$ blocks via a wire cutting method. The thermoelectric module was manufactured by bonding the blocks and a Cu electrode produced on alumina ceramic using Sn–3.0 wt.% Ag–0.5 wt.% Cu alloy solder paste at 260 °C for 6 min. The bonding strength of the thermoelectric module was analyzed by measuring the maximum shear load at breaking using a ball-shear tester (Nordson Corporation, Dage 4000, Westlake, OH, USA). The bonding strength of the thermoelectric module was measured five times and an average value was calculated.

The surface roughness of the thermoelectric element after sand blasting was measured using a laser scanning confocal microscope (LSM700, Carl Zeiss, Oberkochen, Germany) and the bonding interface of the thermoelectric module was observed using a field-emission electron probe microanalyzer (FE-EPMA, JEOL, JXA8530F, Tokyo, Japan). Furthermore, the effect of electroless Ni–P plating layer on the spreadability of the Sn–Ag–Cu-based solder was investigated. Sn–3.0 wt.% Ag–0.5 wt.% Cu alloy solder balls with diameters of 0.76 mm were placed on the surface of thermoelectric elements and maintained at a temperature of 260 °C for 6 min, similar to the bonding conditions of the thermoelectric module. The contact angle between the solder and surface of the thermoelectric element was measured to analyze the spreadability of the solder.

## 3. Results and Discussion

The surfaces of the $3 \times 3 \times 3$ mm$^3$ thermoelectric element blocks were observed using a digital microscope after electroless Ni plating and the results are shown in Figure 1. In the case of the thermoelectric element block prepared without sand blasting, the Ni–P plating layer on the surface was easily delaminated (Figure 1a). However, delamination of the plating layer on thermoelectric element blocks subjected to sand blasting was not observed, even after cutting with a wire saw (Figure 1b–d). The surface of the thermoelectric elements became rougher with increasing particle size of the alumina powder. As there was no chemical bonding at the interface between the thermoelectric element and the Ni–P plating layer, the adhesion of the electroless Ni–P plating layer could be secured via the physical surface roughness effect (i.e., the anchor effect) [23].

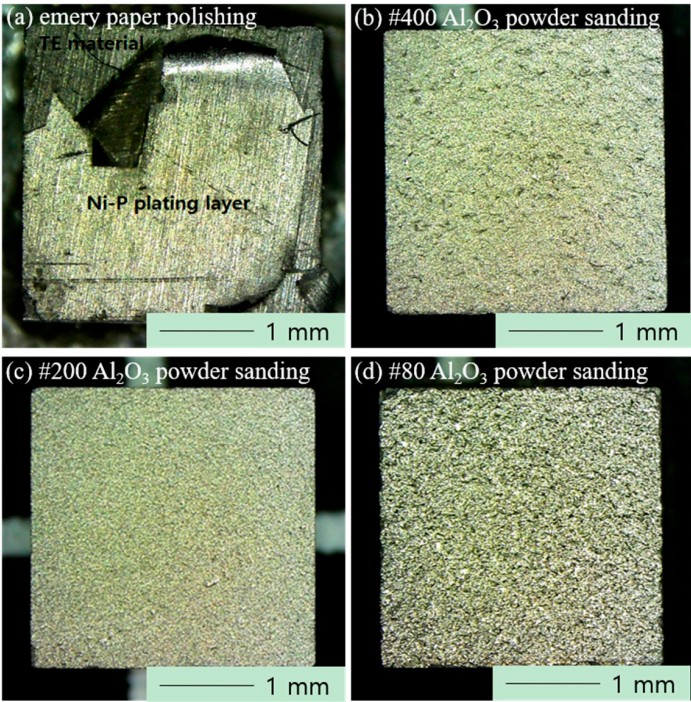

**Figure 1.** Digital microscope images on the surfaces of the thermoelectric element blocks after electroless Ni plating. The images of the surfaces (**a**) polished by emery paper; (**b**) by #400 Al$_2$O$_3$ powder sanding; (**c**) by #200 Al$_2$O$_3$ powder sanding; and (**d**) by the #80 Al$_2$O$_3$ powder sanding.

After sand blasting, the surfaces of thermoelectric element blocks subjected to electroless Ni plating were observed using a laser scanning confocal microscope (Figure 2). As shown in LSCM images, defects such as cracks and pores were not present in the electroless Ni–P plating layer, and the plating layer on top of the thermoelectric element surfaces were uniform. Moreover, LSCM images after electroless Ni–P plating confirm that the arithmetic mean value ($R_a$) of the surface roughness increased with increasing particle size of the alumina powder. This observation shows that the surface roughness of the thermoelectric element could be maintained even after electroless Ni–P plating.

The effect of sand blasting and electroless Ni–P plating on the bonding strength of the Bi–Te-based thermoelectric elements is shown in Figure 3. In the case of the Ni–P-plated specimen prepared without sand blasting, the bonding strength could not be measured as the Ni–P plating layer was easily delaminated. When only sand blasting was applied, the bonding strengths of both the n-type and p-type elements were found to be relatively low, ranging from 4 to 6 MPa. For thermoelectric elements treated with sand blasting (but not Ni–P plating), the bonding strength exhibited no significant change according to the surface roughness. This indicates that the surface roughness of the thermoelectric element has a negligible effect on the bonding strength. However, the bonding strength of the

thermoelectric element treated with both sand blasting and Ni–P plating was higher than that of the specimen treated with only sand blasting. Thus, the bonding strength of the thermoelectric module was improved by applying electroless Ni–P plating. In the case of the electroless Ni-plated specimen, the bonding strength of the thermoelectric module improved with increasing surface roughness of the thermoelectric element. The same tendency was observed in both the n-type and p-type elements.

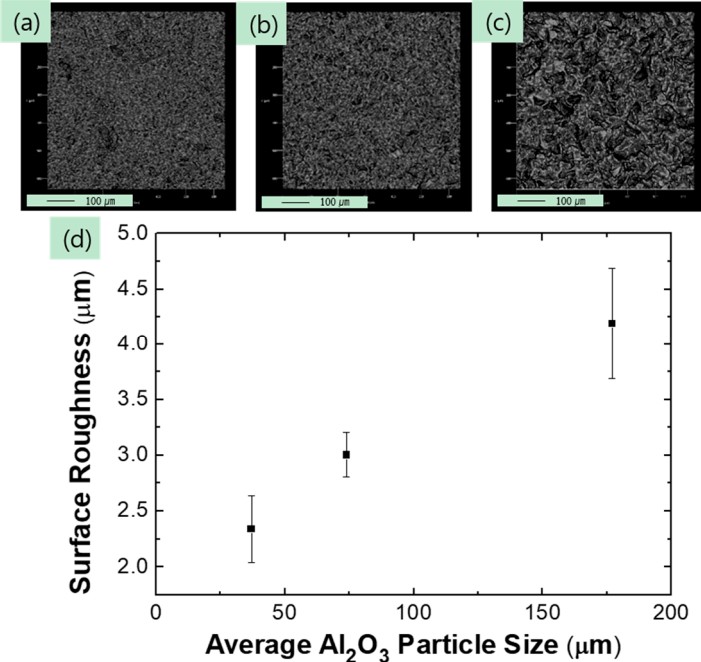

**Figure 2.** Laser scanning confocal microscopy images on the surfaces of thermoelectric element blocks after (**a**) #400 $Al_2O_3$ powder sanding (37 μm); (**b**) #200 $Al_2O_3$ powder sanding (74 μm); and (**c**) #80 $Al_2O_3$ powder sanding (177 μm). (**d**) Surface roughness as a function of alumina powder particle size.

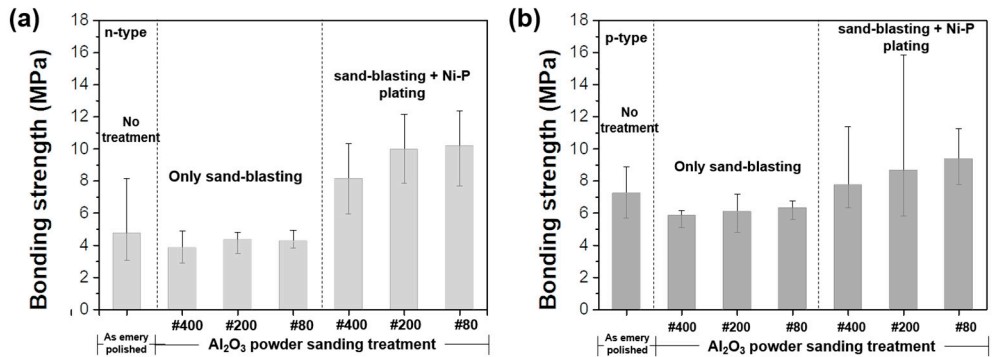

**Figure 3.** Effect of sand blasting and electroless Ni–P plating on bonding strength. Data for Bi–Te-based (**a**) n-type and (**b**) p-type thermoelectric elements.

Figure 4 shows cross-sectional observations and component analysis results obtained using FE-EPMA at the bonding interface between the solder and n-type thermoelectric element without electroless Ni–P plating. The colors in the FE-EPMA analysis show detection signals about each elements. That is, Figure 4b shows the detected amount of only Bi element, and Figure 4c shows the detected amount of only Te element. In the BSE image (Figure 4a), the thermoelectric element and the Sn–Ag–Cu-based solder are bonded, but some pore-like defects exist at the interface. These pores reflect the low spreadability of the molten solder on the surface of the thermoelectric element. Furthermore, the stress applied to the thermoelectric module during the measurement of bonding strength could be concentrated at those defects, resulting in material rupture at low shear stress. As Te

(Figure 4c) and Sn (i.e., the main component of the solder; Figure 4d) were almost evenly distributed at the bonding interface, they became interdiffused during the soldering bonding process and SnTe intermetallic compounds were formed. SnTe intermetallic compounds are known to be brittle [13–17]. Thus, the rupture of the thermoelectric material is attributed to cracks propagating from the SnTe intermetallic compounds.

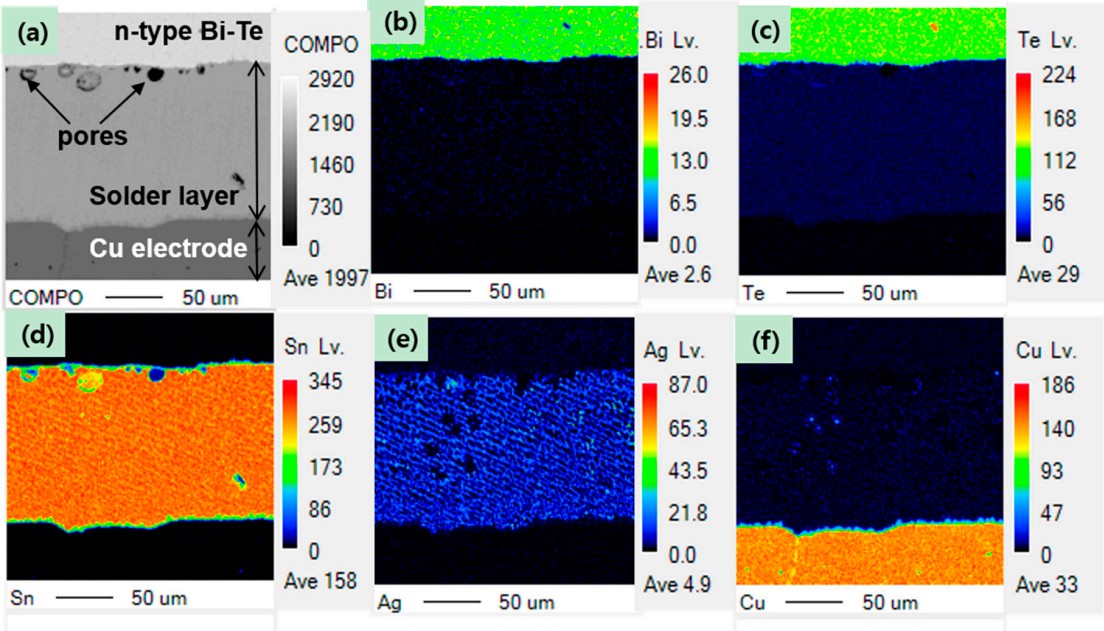

**Figure 4.** Component analysis and cross-sectional observations of the bonding interface between solder and n-type thermoelectric element without electroless Ni–P plating using field emission electron probe microanalysis (FE-EPMA). (**a**) BSE image; (**b**) Bi element map; (**c**) Te element map; (**d**) Sn element map; (**e**) Ag element map; and (**f**) Cu element map.

The cross-sectional observation and component analysis performed using FE-EPMA at the bonding interface between the solder and n-type thermoelectric element with electroless Ni–P plating are shown in Figure 5. No pore-like defects were observed at the bonding interface of the electroless Ni–P plating specimen. As Ni (Figure 5d) was detected between the thermoelectric element and solder bonding, the electroless Ni–P plating layer formed with uniform thickness at the bonding interface.

The concentrations of Bi (Figure 5b) and Te (Figure 5c) below the electroless Ni–P plating layer were very low, suggesting that the plating layer acted as a diffusion barrier, preventing interdiffusion of the thermoelectric element and solder layer. The higher Cu concentrations and lower Sn concentrations observed below the electroless Ni–P plating layer as compared with those of the surroundings indicate the formation of Cu–Sn-based intermetallic compounds. Our FE-EPMA observations of the bonding interface of the p-type thermoelectric modules are consistent with previous studies, which concluded that Cu–Sn-based intermetallic compounds exist in the form of $Cu_3Sn$ and $Cu_6Sn_5$ [23–26], and that they prevent interdiffusion between the Ni–P plating layer and Sn.

A Sn–Ag–Cu solder ball with a diameter of 0.76 mm was melted at 260 °C for 6 min and the cross-sectional view is shown in Figure 6. For the specimen subjected to electroless Ni–P plating, the height of the solder ball was lower than that of the specimen without electroless Ni–P plating, indicating better spreadability. The contact angle between the surface of the thermoelectric element and the solder ball also decreased from 146° to 60° for electroless Ni–P plating. This shows that the wettability of the surface of the electroless Ni–P plating layer with the solder was better than that of the surface of the thermoelectric element alone. This is because Ni has a higher diffusion rate to the

molten solder layer than does Te, resulting in excellent spreadability and wettability and improved bonding strength as the solder diffusion layer thickens.

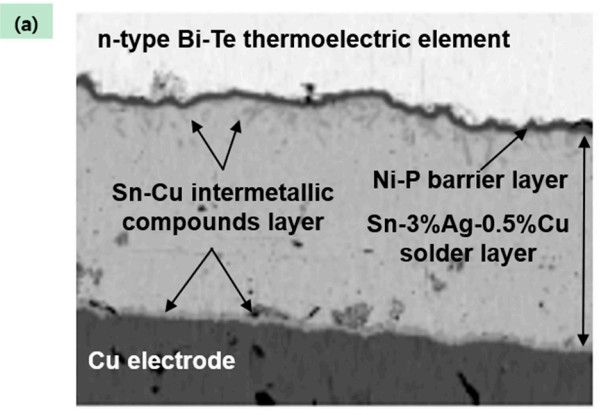

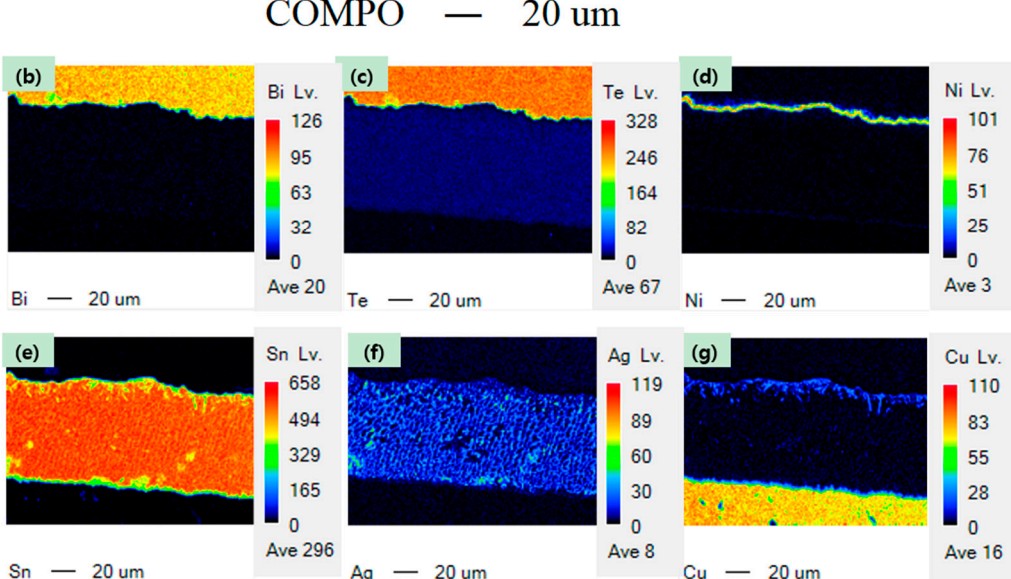

**Figure 5.** Bonding interface between the solder and n-type thermoelectric element with electroless Ni–P plating observed using field emission electron probe microanalysis (FE-EPMA). (**a**) BSE image; (**b**) Bi element map; (**c**) Te element map; (**d**) Ni element map; (**e**) Sn element map; (**f**) Ag element map; and (**g**) Cu element map.

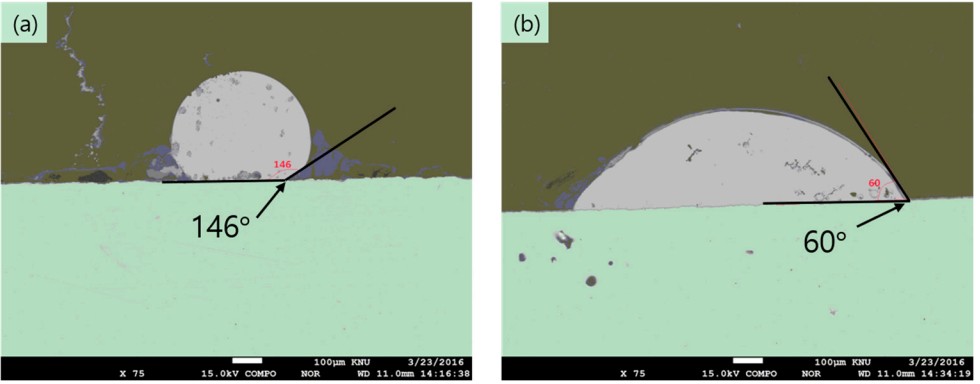

**Figure 6.** Effect of electroless Ni–P plating on solder spreadability of thermoelectric elements. (**a**) without and (**b**) with electroless Ni–P plating.

## 4. Conclusions

In the present study, the effects of an Ni–P plating layer and surface roughness on the bonding strength of a Bi–Te-based thermoelectric module were investigated. Without electroless Ni–P plating, the effect of surface roughness on the bonding strength was negligible. However, the bonding strength was higher when the specimen was subjected to Ni–P plating compared with that without Ni–P plating, and it improved with increasing surface roughness. Brittle SnTe intermetallic compounds were formed at the bonding interface of the thermoelectric element without electroless plating, and defects such as pores were observed in the vicinity. The electroless Ni–P plating layer acted as an effective diffusion barrier layer, suppressing the formation of brittle SnTe intermetallic compounds via interdiffusion of the thermoelectric element and solder. Thus, no pore-like defects were observed at the interface. As electroless Ni–P plating improved the wettability with molten solder, an increase in bonding strength was attributed to the formation of a thicker solder reaction layer below the bonding interface. In the case of the specimen subjected to electroless Ni–P plating, the bonding strength improved owing to the increase in the bonding interface with the solder at higher surface roughness.

**Author Contributions:** Formal Analysis, S.H.Y. and H.C.; Investigation, S.K., I.S. and K.T.K.; Writing–Original Draft Preparation, S.H.B.; Writing–Review & Editing, S.H.B., S.K., S.H.Y., I.S. and K.T.K.

**Funding:** This research was funded by the National Research Foundation of Korea (No. NRF-2017R1D1A1B03030792).

**Conflicts of Interest:** The authors declare no competing interests.

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
