# Peer review of "Effect of Surface Roughness and Electroless Ni–P Plating on the Bonding Strength of Bi–Te-based Thermoelectric Modules"

_coatings, doi:10.3390/coatings9030213_

Reviewer 1 Report

The manuscript "Effect of Surface Roughness and Electroless Ni-P Plating on the Bonding Strength of Bi-Te-based Thermoelectric Modules" by Bae, Son, Kim and Chung, relates an interesting study focusing on the interfacial structure properties of thermoelectric modules and the electrodes that are required to use them as energy generators. The structural robustness of the electrode contact with the thermoelectric material (and its electrical conductivity) is an important topic in the field of thermoelectric materials development, so the study is quite relevant to the field and reveals some new insight into techniques for improving the electrode attachment. I have a few comments that focus on improving the clarity of the paper as outlined below.

1. On line 67 the authors state that "Three kinds of powders..." were used. While some readers may infer that this is talking about the silicon oxide mentioned in the previous section, it will be more clear to re-state what kind of powder is used here. 

2. The descriptions for Figure 1 and 2 as well as the figure captions is a bit confusing. The Figure 1 caption states that the images are for thermoelectric blocks after plating while the Figure 2 caption states that the images are for thermoelectric blocks after sand blasting. The in-text descriptions indicates that all images are taken after plating. These descriptions should be cleaned up to be consistent. 

3. Figure 4 and 5 show FE-EPMA elemental maps, however, some images show more than one color in the map. For example, Figure 4C shows a bright yellow at the top and a blue color in the middle. An explanation of the meaning of the colors would clarify this.

Author Response

Reviewer #1 General Comment : The manuscript "Effect of Surface Roughness and Electroless Ni-P Plating on the Bonding Strength of Bi-Te-based Thermoelectric Modules" by Bae, Son, Kim and Chung, relates an interesting study focusing on the interfacial structure properties of thermoelectric modules and the electrodes that are required to use them as energy generators. The structural robustness of the electrode contact with the thermoelectric material (and its electrical conductivity) is an important topic in the field of thermoelectric materials development, so the study is quite relevant to the field and reveals some new insight into techniques for improving the electrode attachment. I have a few comments that focus on improving the clarity of the paper as outlined below.

Comment 1: On line 67 the authors state that "Three kinds of powders..." were used. While some readers may infer that this is talking about the silicon oxide mentioned in the previous section, it will be more clear to re-state what kind of powder is used here.

Response : As you commented, we have added “commercial grade Al2O3 ” in the Experimental methods of the revised manuscript, as follows. “Three kinds of commercial grade Al2O3 powders—#80 (mean particles size: 177 µm), #200 (mean particles size: 74 µm), and #400 (mean particles size: 37 µm)—were sprayed onto the disk surface using the sand blasting method to prepare thermoelectric elements with different surface roughness characteristics.”

Comment 2 : The descriptions for Figure 1 and 2 as well as the figure captions is a bit confusing. The Figure 1 caption states that the images are for thermoelectric blocks after plating while the Figure 2 caption states that the images are for thermoelectric blocks after sand blasting. The in-text descriptions indicates that all images are taken after plating. These descriptions should be cleaned up to be consistent.

Response : We have corrected the captions of both Figures 1 and 2 to clarify its description, as follows. The caption of Figure 1 is corrected as “Digital microscope images on the surfaces of the thermoelectric element blocks after electroless Ni plating. The images of the surfaces (a) polished by emery paper; (b) by #400 Al2O3 powder sanding; (c) by #200 Al2O3 powder sanding; and (d) by the #80 Al2O3 powder sanding.” And following statements are added in the revised manuscript. “After sand blasting, the surfaces of thermoelectric element blocks subjected to electroless Ni plating were observed using a laser scanning confocal microscope (Figure 2). As shown in LSCM images, defects such as cracks and pores were not present in the electroless Ni-P plating layer, and the plating layer on top of the thermoelectric element surfaces were uniform. Moreover, LSCM images after electroless Ni–P plating confirm that the arithmetic mean value (Ra) of the surface roughness increased with increasing particle size of the alumina powder. This observation shows that the surface roughness of the thermoelectric element could be maintained even after electroless Ni–P plating. “ According to the comments, we have added the corrected Figure 2 in the revised manuscript. The caption of Figure 2 is corrected into “Laser scanning confocal microscopy images on the surfaces of thermoelectric element blocks after (a) #400 Al2O3 powder sanding (37 µm); (b) #200 Al2O3 powder sanding (74 µm); and (c) #80 Al2O3 powder sanding (177 µm), (d) Surface roughness as a function of alumina powder particle size.”

Comment 3: Figures 4 and 5 show FE-EPMA elemental maps, however, some images show more than one color in the map. For example, Figure 4C shows a bright yellow at the top and a blue color in the middle. An explanation of the meaning of the colors would clarify this.

Response : We agree with your comments on the Figure 4 and 5. The colors in FE-EPMA analysis shows detection signals about each elements. That is, Figure 4b shows the detected amount of only Bi element and Figure 4c shows the detected amount of only Te element. The intensity is expressed as colors. Thus, we added the values in FE-EPMA mapping images for more clear comparison in both Figures. And we added the following statement in the main text of the revised manuscript. “The colors in FE-EPMA analysis shows detection signals about each elements. That is, Figure 4b shows the detected amount of only Bi element and Figure 4c shows the detected amount of only Te element.”

Reviewer 2 Report

It is a great article dealing with the effect of surface roughness and electroless Ni–P plating on the bonding strength of Bi–Te-based thermoelectric modules.

The article is well built and the results are scientifically significant.

I have several recommendations for gaining more interest to the paper by a broad range of researchers, dealing with the application of bismuth telluride based modules.

For those, I would recommend to include in the article the following references:

O. Beeri, O. Rotem, E. Hazan, E.A. Katz, A. Braun and Y. Gelbstein, Hybrid photovoltaic-thermoelectric system for concentrated solar energy conversion: experimental realization and modeling, Journal of Applied Physics 118(11) 115104 (2015).

R. Vizel, T. Bargig, O. Beeri and Y. Gelbstein, Bonding of Bi2Te3- based thermoelectric legs to metallic contacts using Bi0.82Sb0.18 alloy, Journal of Electronic Materials 45(3) 1296-1300 (2016).

Dealing with a hybrid photovoltaic-thermoelectric application and the bonding of such elements into couples, respectively.

Following taking into accounts the minor revisions specified above I will be glad to recommend on acceptance of the manuscript.

Author Response

General Comment :  It is a great article dealing with the effect of surface roughness and electroless Ni–P plating on the bonding strength of Bi–Te-based thermoelectric modules.The article is well built and the results are scientifically significant.I have several recommendations for gaining more interest to the paper by a broad range of researchers, dealing with the application of bismuth telluride based modules.

Comment 1 : For those, I would recommend to include in the article the following references:

O. Beeri, O. Rotem, E. Hazan, E.A. Katz, A. Braun and Y. Gelbstein, Hybrid photovoltaic-thermoelectric system for concentrated solar energy conversion: experimental realization and modeling, Journal of Applied Physics 118(11) 115104 (2015). R. Vizel, T. Bargig, O. Beeri and Y. Gelbstein, Bonding of Bi2Te3- based thermoelectric legs to metallic contacts using Bi0.82Sb0.18 alloy, Journal of Electronic Materials 45(3) 1296-1300 (2016).

Dealing with a hybrid photovoltaic-thermoelectric application and the bonding of such elements into couples, respectively. Following taking into accounts the minor revisions specified above I will be glad to recommend on acceptance of the manuscript.

Response : Thank you for your kind review. As you recommended, we added your two papers in the Reference. [25] O. Beeri, O. Rotem, E. Hazan, E.A. Katz, A. Braun and Y. Gelbstein, Hybrid photovoltaic-thermoelectric system for concentrated solar energy conversion: experimental realization and modeling, Journal of Applied Physics 118(11) (2015) 115104. [26] R. Vizel, T. Bargig, O. Beeri and Y. Gelbstein, Bonding of Bi2Te3- based thermoelectric legs to metallic contacts using Bi0.82Sb0.18 alloy, Journal of Electronic Materials 45(3) (2016) 1296-1300.

Reviewer 3 Report

The authors examined the bonding strength of thermoelectric modules in dependence of the roughness of the surface with and without Ni-P plating.

The results and conclusions are rather vage if one has a deeper look to the huge standard deviations (see Figure3). Thus the effect of sand - blasting and Ni-P plating is not well proven and the conclusion that this improves the thermoelectric module stability is rather vage.

Despite this criticism the units should be given in correct dimensions and the parts of figure 4 and 5 should have the same dimensions.

Nevertheless it is an interesting concept to improve the quality of thermoelectric modules.

Author Response

Comment 1 : The authors examined the bonding strength of thermoelectric modules in dependence of the roughness of the surface with and without Ni-P plating.

The results and conclusions are rather vage if one has a deeper look to the huge standard deviations (see Figure 3). Thus the effect of sand - blasting and Ni-P plating is not well proven and the conclusion that this improves the thermoelectric module stability is rather vage.

Response : Thank you for your kind review. Although the bonding strength increase with increasing the surface roughness was not perfectly shown in this experiment, it was clearly confirmed that the average values of bonding strength tend to increase as a function of surface roughness as shown in Figure 3. Therefore, we can conclude that the bonding strength of Bi-Te thermoelectric module after Ni-P electroless plating is mainly influenced by increasing surface roughness.

Comment 2: Despite this criticism the units should be given in correct dimensions and the parts of Figures 4 and 5 should have the same dimensions.

Response : according to your comment, we have added the detection signal values in FE-EPMA mapping images of Figures 4 and 5 in the revised manuscript.

Round  2

Reviewer 3 Report

I recommend minor revision